# A Novel Stick-Slip Nanopositioning Stage Integrated with a Flexure Hinge-Based Friction Force Adjusting Structure

**DOI:** 10.3390/mi11080765

**Published:** 2020-08-11

**Authors:** Junhui Zhu, Peng Pan, Yong Wang, Sen Gu, Rongan Zhai, Ming Pang, Xinyu Liu, Changhai Ru

**Affiliations:** 1School of Mechatronic Engineering and Automation, Shanghai University, Shanghai 200444, China; juviechu@shu.edu.cn (J.Z.); 15106201197@163.com (Y.W.); zra123@shu.edu.cn (R.Z.); 2Department of Mechanical and Industrial Engineering, University of Toronto, Toronto, ON M5S 3G8, Canada; peng.pan@mail.mcgill.ca (P.P.); xyliu@mie.utoronto.ca (X.L.); 3Department of Mechanical Engineering, McGill University, Montreal, QC H3A 0C3, Canada; 4College of Mechanical and Electrical Engineering & Robotics and Microsystem Center, Soochow University, Suzhou 215021, China; hwigusen2014@163.com; 5College of Automation, Harbin Engineering University, Harbin 150001, China; pangm@hrbeu.edu.cn; 6Micro-Nano Automation Institute, JITRI, Suzhou 215100, China

**Keywords:** piezoelectric actuation, stick-slip, nanopositioning, friction force, flexure hinge

## Abstract

The piezoelectrically-actuated stick-slip nanopositioning stage (PASSNS) has been applied extensively, and many designs of PASSNSs have been developed. The friction force between the stick-slip surfaces plays a critical role in successful movement of the stage, which influences the load capacity, dynamic performance, and positioning accuracy of the PASSNS. Toward solving the influence problems of friction force, this paper presents a novel stick-slip nanopositioning stage where the flexure hinge-based friction force adjusting unit was employed. Numerical analysis was conducted to estimate the static performance of the stage, a dynamic model was established, and simulation analysis was performed to study the dynamic performance of the stage. Further, a prototype was manufactured and a series of experiments were carried out to test the performance of the stage. The results show that the maximum forward and backward movement speeds of the stage are 1 and 0.7 mm/s, respectively, and the minimum forward and backward step displacements are approximately 11 and 12 nm, respectively. Compared to the step displacement under no working load, the forward and backward step displacements only increase by 6% and 8% with a working load of 20 g, respectively. And the load capacity of the PASSNS in the vertical direction is about 72 g. The experimental results confirm the feasibility of the proposed stage, and high accuracy, high speed, and good robustness to varying loads were achieved. These results demonstrate the great potential of the developed stage in many nanopositioning applications.

## 1. Introduction

A positioning stage with nanometer resolution is widely used in many fields such as bioengineering, nanomanipulation, and nanometrology [1,2,3,4,5]. In order to obtain nanometer resolution, a piezoelectric actuator is often employed as the actuator of the positioning stage for its significant performance, such as high frequency, nanometer resolution, and high energy density [6]. However, the travel range of a piezoelectric actuator is quite small, which is only in the micrometer range. To meet the demand of long travel range and nanometer resolution at the same time, many techniques have been developed to amplify or accumulate the displacement of the piezoelectric actuator, including the macro-micro dual drive principle [7,8], the inchworm drive principle [9], a piezoelectric ultrasonic motor [10,11], and the stick-slip drive principle [12,13]. Stages based on the macro-micro dual drive principle usually have large dimensions, making them unable to be used in many applications of limited space [14]. The size of an inchworm drive stage is also large, and it needs to be manufactured and assembled precisely [15]. As for the piezoelectric ultrasonic motor, its driving force is small, which limits its use in applications where heavy loads are required [16]. Compared with them, stages developed by the stick-slip drive principle not only have a compact structure, but also provide high resolution and long travel range [17], which enables this kind of stage to be commonly employed in the scanning electron microscope (SEM). Owing to its inherent excellent characteristics, the piezoelectrically-actuated stick-slip nanopositioning stage (PASSNS) has been a hot research topic.

The stick-slip principle inertial motor was originally investigated by Yoshida et al. [18] and firstly commercialized by Konica–Minolta, Japan [19]. They revealed the principle of the “stick & slip” motion. The essence of the stick-slip principle is the friction force change between maximum static friction force and sliding friction force [20,21]. As illustrated in Figure 1, the piezoelectric actuator pushes the driving object forward and the slider keeps relatively static to the driving object when the driving signal rises slowly from O to A. If the driving signal descends quickly from A to B, the driving object will retract fast. This will result in a relative movement between driving object and slider, and the slider almost remains static to the ground. In a circle of the driving signal, the slider will get a forward displacement. If a continuous driving signal is applied to the piezoelectric actuator, the slider will produce a large travel range. Obviously, the friction force is of quite importance for the successful movement of a PASSNS.

Until now, many designs of PASSNSs have been developed. Zhang et al. utilized the stick-slip principle to develop a stage including a horizontal movement and a rotary movement [22]. This stage has a maximum speed of 7.2 mm/s and a maximum output force of 2.09 N. Rakotondrabe et al. developed a stick-slip stage which is capable of moving at a maximum speed at 1.8 mm/s in the horizontal direction, and has a step resolution of 70 nm and an angular resolution of 0.001° [23]. A systematic investigation and performance comparisons of different stick-slip and slip-slip modes of operation were previously discussed in Hunstig M.’s research [24]. Criteria such as steady state velocity, smoothness of motion, and start-up time were used for comparisons. Shimizu et al. utilized the stick-slip principle and piezoelectric actuators to develop an XY positioning stage [25]. The stage was capable of moving over a range of ±1 mm in both directions at a travel speed of 5 mm/s. In addition to the extensive research on PASSNSs by scholars, some companies have issued related commercialized products. For example, the 1-DOF linear positioning stage (Model SLC-1720), a classic product of SmarAct GmbH, is designed based on the stick-slip driving principle; its maximum stroke is 12 mm, the maximum resolution is 50 nm, the speed is more than 20 mm/s, the weight is only 13 g, and the size is 22 × 17 × 8.5 mm^3^ [26]. The most representative 1-DOF stick-slip drive positioning stage of Attocube System Inc. is the ANPx101 [27] with a size of 24 × 24 × 11 mm^3^, a weight of 20 g, a position sensor resolution of about 200 nm, a stroke of 5 mm, a speed that can reach 3 mm/s, and a maximum load that can reach 1 N. The main characteristics of the proposed stages of these studies are summarized in Table 1.

The PASSNSs reported in these literatures and some commercialized products research focuses are different, they are more about the structural design and positioning performance of the stage itself. There are literature reports on friction adjustment mechanisms and methods [28,29,30], such as mechanical tolerance fit, permanent magnet preloading, and upper and lower springs preloading, but these friction adjustment methods are more complex, which will restrict the friction adjustment. At the same time, these friction adjustment methods have higher requirements on the machining accuracy of the friction interface, which leads to higher requirements on the machining and assembly accuracy of the PASSNS, and ultimately affects the positioning performance of the stage. Although many PASSNSs have been developed, there still exist many problems such as low working load, step displacement sensitive to the load change, hard to obtain vertical movement, poor dynamic performance, and so on.

In order to realize the movement positioning with nanoscale positioning performance and strong robustness to load change, a novel PASSNS integrated with a flexure hinge-based friction force adjusting structure was developed. In addition, we built a general dynamic model and performed a simulation to analyze the influence of various parameters on the performance of the stage. Finally, experiments were conducted on a prototype of the developed PASSNS. High accuracy, high speed, large load capacity, and large travel range were achieved, which demonstrates that this stage has significant performance and potential in many applications.

**Figure 1 micromachines-11-00765-f001:**
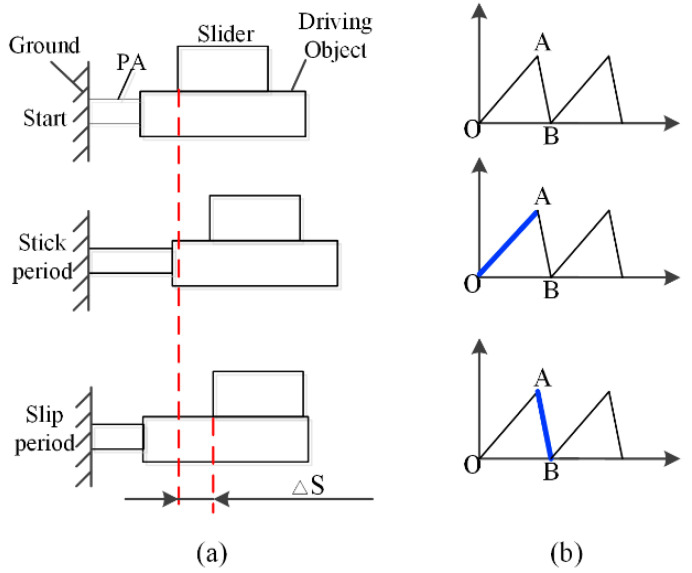
(**a**) Principle of stick-slip driving; (**b**) conventional driving signal. Reproduced with permission from [31].

## 2. Design and Analysis

### 2.1. Design of the Piezoelectrically-Actuated Stick-Slip Nanopositioning Stage (PASSNS)

The structure of the designed PASSNS is illustrated in Figure 2. It can be clearly seen that the stage mainly consists of a driving unit, a moving unit, and an adjusting unit. In this design, the cross roller guide rail is recognized as the slider. The driving unit includes a piezoelectric actuator and flexure hinges A; the unit is used to preload. The moving unit includes flexure hinges B; the unit is used to transfer the displacement of the actuator, a mass block, and a set of cross roller guide rails. The adjusting unit has a flexure hinge structure linked to the mass block and one bolt to change deformation of the flexure hinges above the mass block. This flexure hinge structure possesses a friction block and flexure hinges C. Friction force between the friction block and the guide rail can be adjusted by screwing the bolt to change the deformation of the flexure hinges C. This bolt goes through the threaded hole manufactured at the center of the mass block to ensure that supporting force is applied to the center of the flexure hinges C. As the deformation of the flexure hinge is a result of the distance that the bolt goes inside, it will not change when the working load changes. In addition, the working load compared to the positive pressure between friction block and guide rail is very small, thus when the load changes, the step displacement will not change. This provides the developed PASSNS an ability to be robust to varying load.

### 2.2. Design of Friction Force Adjustment Structure

As mentioned above, friction force is the key to successful movement of the PASSNS. To enable the stick-slip nanopositioning stage to move in both horizontal and vertical directions, it is necessary to adjust the friction force between the driving object and the slider. As the flexure hinge provides many advantages over conventional joints such as being monolithic with the rest of the mechanism, there is no need for lubrication, no backlash, and compactness [32]; it is employed to uniformly adjust friction force between the driving object and slider with a bolt in contact with its center. A schematic of the flexure hinge structure C is shown in Figure 2; it is used to adjust friction force, as shown in Figure 3. This structure consists of two symmetric corner-filleted flexure hinges. When the bolt for adjusting friction force is screwed upwards, flexure hinges C will deform symmetrically as shown in Figure 3b. Finally, the friction force between these two parts will increase accordingly. The flexure hinge-based friction force adjusting structure is made by integrated manufacturing, the structure and adjustment method are simple, easy to manufacture and assemble, and the assembly error is reduced.

### 2.3. Determination of Flexure Hinge Size

When designing the flexure hinge, the effect of the flexure hinge’s stiffness on the resonance frequency of the PASSNS should be taken into consideration. If the stiffness of the flexure hinge is large, the dynamic performance of the stage will be good. However, if the stiffness is too large, the flexure hinge under a small deformation will endure a large stress in the root of the flexure hinge. To avoid structure failure and poor dynamic performance of the stage, stiffness of the stage should be reasonable. Stiffness of the flexure hinge structure along vertical direction is described as *K* which is calculated as:(1)$$K=\frac{F}{w}=\frac{2th^{3}E}{L^{3}}$$
where *t* is the width of the single corner-filleted flexure hinge, *L* is the length, and *h* is the thickness of the single flexure hinge. *E* is the elastic modulus of material. To obtain proper dimensions of flexure hinges, a flow chart has been developed, shown in Figure 4. As 7075-T651 aluminum (Young’s modulus value: 71 × 109 N/m^2^; Poisson’s ratio: 0.33) has many advantages like small density, good abrasion performance, and high strength, it was used as the material to manufacture the proposed PASSNS. To conduct finite element analysis (FEA) by ANSYS, the 10-node solid 92 tetrahedron element was used. Finally, the dimensions of the flexure hinge B were set to be 4 mm long, 4 mm wide and 0.6 mm thick and the flexure of C was designed to be 1.5 mm (length) × 4 mm (width) × 0.4 mm (thickness). The stiffness of flexure hinge B was calculated by the formula and was 3.888 N/μm; the stiffness obtained from the FEA was 3.777 N/μm. The error between theoretical stiffness and simulation stiffness was 2.83%, which proved validity of the stiffness formula. In addition, when the flexure hinge experienced 10 μm deformation, maximum stress in the root of the flexure hinge was calculated to be 36.2 Mpa, which is far less than the yield stress of 7075-T651 aluminum of 505 Mpa. The stiffness of the flexure hinge C was calculated to be 10.771 N/μm. When a 100 N force was applied to the flexure hinge C, maximum stress was calculated to be 351.5 Mpa, which is also less than the yield stress of the 7075-T651 aluminum. According to the FEA results, we obtained first order natural frequency of this stage base which was 10 kHz, as is shown in Figure 5. Thus, the frequency of the driving signal applied to the piezoelectric actuator should be below 10 kHz. The main characteristic parameters of the A, B, C hinges are summarized in Table 2.

### 2.4. Analysis and Simulation of the PASSNS

To study the effect of different parameters on the performance of the PASSNS, it is necessary to establish the dynamic model. As shown in Figure 1, in the driving phase, the piezo stack actuator outputs displacement when the driving signal is applied. Then, the displacement is transferred to the mass block through the flexure hinge B. Finally, forward/backward motion of the moving guide rail will be obtained through the switch of dynamic and static friction force between the mass block and guide rails. Thus, the whole dynamic model of the stage can be divided into three separate models, which are the electric model of the piezo stack actuator, the driving and transmission model of piezo stack actuator and mass block, and the stick-slip transmission model between the mass block and guide rails, as shown in Figure 6. If there exists a tilting angle *θ*, gravity of the moving guide rail has a large influence on the movement of the stage and it should be considered. In the whole dynamic model, the piezo stack actuator is placed on the left side of the mass block. If the signal in Figure 1b is applied to the piezo stack actuator, the moving guide rail may move forward continuously. To describe the friction force precisely, the LuGre friction model [33] has been used. After analysis of each model, we obtain the transfer functions of each model. Based on the transfer functions of each model, we have established a simulation model, presented in Figure 7, using MATLAB.

Using the simulation model, the effects of some parameters on the performance of the PASSNS’s movement can be studied. The systematic study of the PASSNS is mainly performed in the case that the stage is placed horizontally. In this case, the tilting angle is 0 degree. At first, the parameter of total mass of the mass block and adjusting unit is studied, and the simulation result is shown in the Figure 8. From this simulation result, we can see that the average step displacements are about 1.55 and 1.3 μm, respectively, when the total mass of the adjusting unit and mass block are 0.025 and 0.25 kg, respectively, with a decrease rate of about 16.13%. In addition, the displacement curve of the mass block vibrates obviously from the time between 0.021 and 0.028 s, and between 0.043 and 0.048 s, when the total mass increases to 0.25 kg. To ensure the system is stable and the step displacement is as large as possible, the total mass of the mass block and adjusting unit should be smaller than the mass of the guide rails. In our work, the total mass of the mass block is about 0.855 g and the mass of the guide rails is about 7.1 g. The employment of the flexure hinge as an adjusting unit makes it possible, as the flexure hinges provide many advantages such as being monolithic with the rest of the mechanism, no backlash, and compactness.

As known to all, the movement of the PASSNS relies on friction force. Conventionally, when the working load changes, the friction force between the driving object and the slider will change accordingly. From the simulation results as shown in Figure 9, we can see that in the conventional stage design, the step displacement will change by 17.11% from 3.8 to 4.45 μm with increasing load.

## 3. Experimental Results

A prototype of the proposed PASSNS, as shown in Figure 10a, was fabricated by using electro-discharging machining with a resolution better than 0.5 mm. The piezoelectric actuator (PANT, PTJ1500303051) can provide a maximum displacement of 5 μm under 150 V applied voltage and the maximum output force it provides is 150 N. The cross-roller guide (GMT, GRV01-30) provides a maximum stroke of 21 mm. The material of the stage base is 7075-T651 aluminum and the overall size of this whole stage is 24 mm (length) × 12 mm (height) × 30 mm (width).

The experimental system was established to characterize the performance of the proposed PASSNS, as shown in the Figure 10b. In the experimental setup, an NI PCI-6221 DAQ card integrated into the PC was used to generate signals (e.g., conventional and modified sawtooth). A power amplifier (BOSHI) amplified the signals to drive the piezoelectric actuator. Displacement of the horizontal positioning stage was measured by a Polytec laser vibrometer, as shown in Figure 10b, and displacement of the vertical positioning stage was measured by a noncontact and compact laser distance sensor (KEYENCE, LKG3000 Series), which is mounted on a base placed on an anti-vibration table. The experimental data measured by the Polytec laser vibrometer or KEYENCE were collected into the LabVIEW software of the computer through the NI PCI-6221 DAQ card.

Before testing the performance of the PASSNS, we adjusted the friction force between the friction block and guide rail to enable the stage to move successfully. When the conventional sawtooth driving signal, with the frequency of 6 Hz and amplitude of 80 V, was applied to drive the PASSNS forwards and backwards, displacements of the stage in two directions, respectively, were obtained. After calculation, the average forward and backward step displacements were measured to be 2.27 and 1.36 μm, respectively. The step displacements in opposite directions are different due to the nonuniform roughness of the surface between the guide rails and the friction block. If it can be improved, the step displacements in opposite directions will be closer. To quantify the precision of step displacement, we obtained 20 continuous upward and downward step displacements, respectively, shown in Figure 11. Positioning accuracy for upward and downward step displacements were calculated to be 0.09 and 0.1 μm, respectively. This shows that the PASSNS can provide relatively high positioning accuracy.

In the following experiments, the maximum speed of the PASSNS has been tested with no working load. After a series of experiments, we found when the frequency of the conventional sawtooth driving signal reaches 600 Hz, forward and backward movement of stage is the fastest. The displacement of the stage is plotted in Figure 12; here the amplitude of the applied driving signal is 80 V. Finally, the maximum speeds of the forward and backward movements were measured to be 1.00 and 0.70 mm/s, respectively.

After that, the minimum step displacement of forward and backward movement of the PASSNS was studied. When the amplitude and cycle of the driving signal are set to be 4.5 V and 60 ms (each cycle is divided into two parts, one of which is 35 ms and another one is 25 ms), respectively, the minimum step displacement can be obtained accordingly. As there exists measurement noise, we measured 50 continuous step displacements and calculated the average value with standard deviation of 0.002 μm as the minimum step displacement. As shown in Figure 13a, the total displacement of 50 continuous forward movements is calculated to be 0.55 μm. Thus, after calculation, the minimum forward step displacement is around 11.00 nm. For the backward movement, the minimum step displacement is measured to be 12.00 nm.

As mentioned above, in the conventional stick-slip positioning stage, the step displacement varies drastically when the load changes. Here, we studied the effect of a working load on the the proposed PASSNS’s performance. As the guide rails have weight of around 7.104 g, it is recognized as the initial load. In Figure 11, the step displacement of the stage under initial load has been measured. Here we put the working load of 20 g on the stage and measured its step displacement while other parameters were kept the same. For the forward movement, the average step displacement was calculated to be 2.41 μm, as shown in Figure 14. Compared to the step displacement under initial load, the step displacement only increases by 6% even though the working load is three times larger than the initial load. For the backward movement, the average step displacement was calculated to be 1.47 μm. Compared with the backward step displacement with no load, the backward step displacement only increases by 8% even though the working load is three times larger than the initial load. Thus, compared with the traditional stage design simulation results in Figure 9, our PASSNS design is more robust to the varying load.

The loading capacity of the PASSNS in the vertical direction depends on the friction force between the stick-slip surfaces and slider mass. On the other hand, it depends on the load it can move upward. We carried out the vertical load experiments: we applied a driving signal with a period of 200 ms and an amplitude of 150 V to the piezoelectric ceramic of the positioning stage. When the load is gradually increased in the vertical direction, and when it is increased to 72 g, the positioning stage can move upward and output a rising displacement, as shown in Figure 15a. When the load is increased to more than 72 g, the slider of the positioning stage does not move upward, but falls down, as shown in Figure 15b. Thus, based on the experimental results, the maximum load of the stage in the vertical direction is 72 g.

## 4. Discussion and Conclusions

As known to all, friction force plays an important role in the successful movement of the PASSNS. To effectively change the friction force, many friction force adjusting units have been integrated into the PASSNS. However, most of these units cannot allow the stage to have a robust performance when subjected to varying load. When the working load changes, the performance of the stage will change drastically. In addition, some friction force adjusting units are heavy and large in size, which results in a drastic vibration of the stage during movement. To overcome these problems, we developed a novel PASSNS where the flexure hinge-based friction force adjusting unit was employed to meet requirements. Numerical analysis was conducted to estimate the static performance of the PASSNS. To study the effect of different parameters on the movement performance of the PASSNS, a dynamic model was established and simulation analysis was performed. Based on the simulation analysis, we found that different total masses of the adjusting unit and the mass block will affect the dynamic performance of the stage. We also found that in the conventional stage design, the step displacement of the stage will become large with the increase of the working load. The employment of the flexure hinge-based friction force adjusting unit helps us reduce the weight of the adjusting unit, which improves the dynamic performance of the PASSNS. In addition, the deformation of the flexure hinge can maintain the same position when the load changes. This ensures that the stage can be robust to the varying load. This is further demonstrated by the experimental results. The maximum speed of forward and backward movement of the stage has been measured to be 1.00 and 0.70 mm/s, respectively, when the amplitude of the driving signal was set to be 80 V. The minimum forward step displacement is around 11.00 nm, and the minimum backward step displacement is around 12.00 nm. The maximum vertical direction load capacity of the stage is 72 g with robustness to varying load. The proposed PASSNS provides excellent performance, which demonstrates the great potential of the developed stage in many applications which require nanometer positioning accuracies, high travel stroke, compact structure, and a large load.

## Figures and Tables

**Figure 2 micromachines-11-00765-f002:**
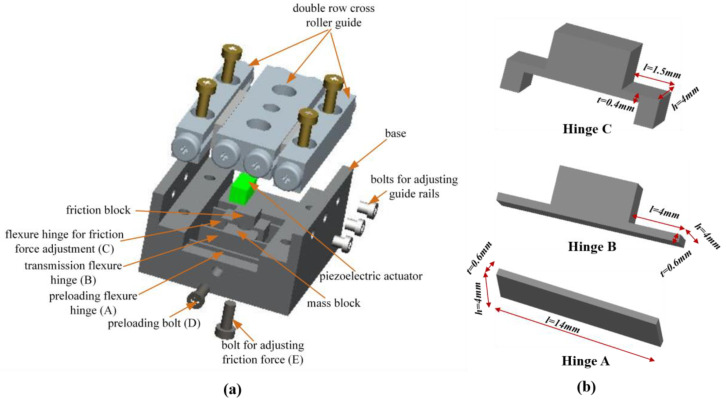
(**a**) Schematic of the developed stick-slip positioning stage. Reproduced with permission from [31]; (**b**) zoom in on the different hinges (A, B, C).

**Figure 3 micromachines-11-00765-f003:**
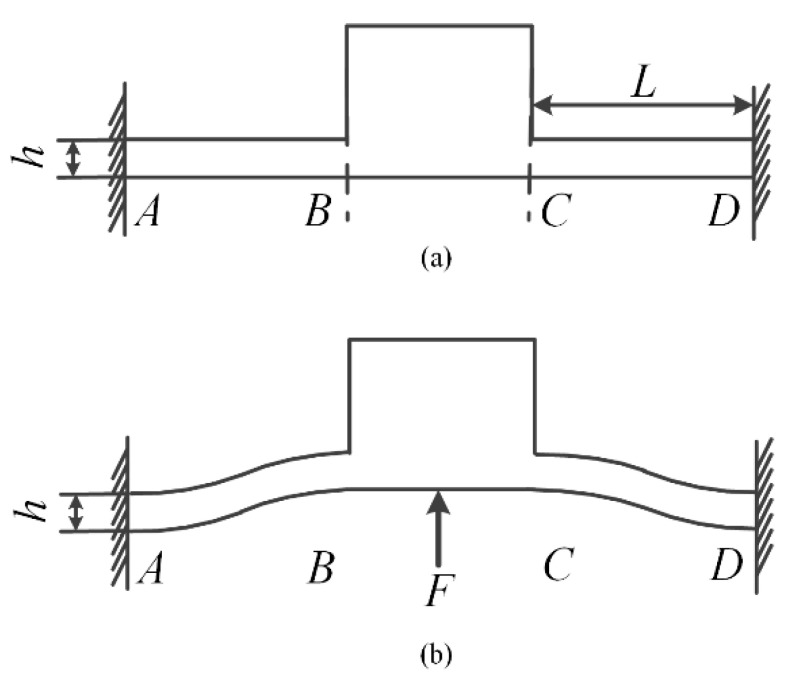
Schematic of the flexure hinges to adjust friction force: (**a**) the structure of flexure hinge; (**b**) force diagram of the flexure hinge. Reproduced with permission from [31].

**Figure 4 micromachines-11-00765-f004:**
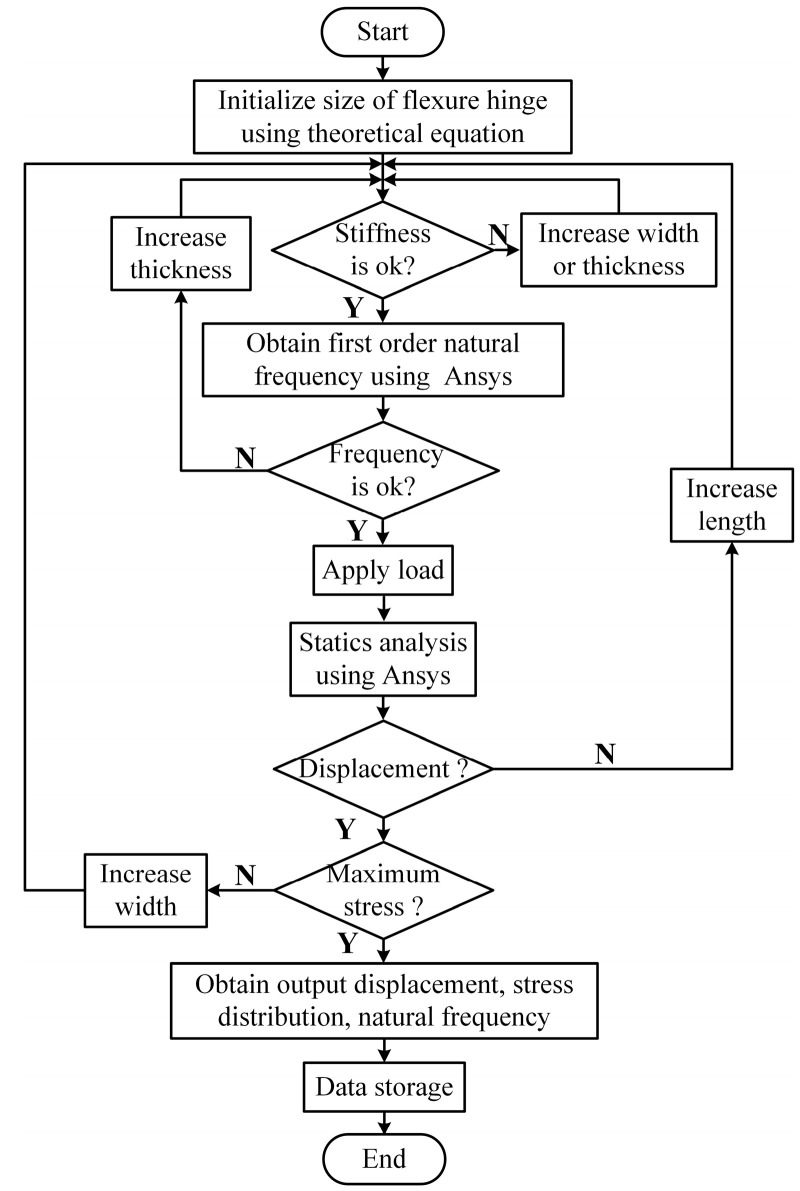
Flow chart to design dimensions of the flexure hinge. Reproduced with permission from [31].

**Figure 5 micromachines-11-00765-f005:**
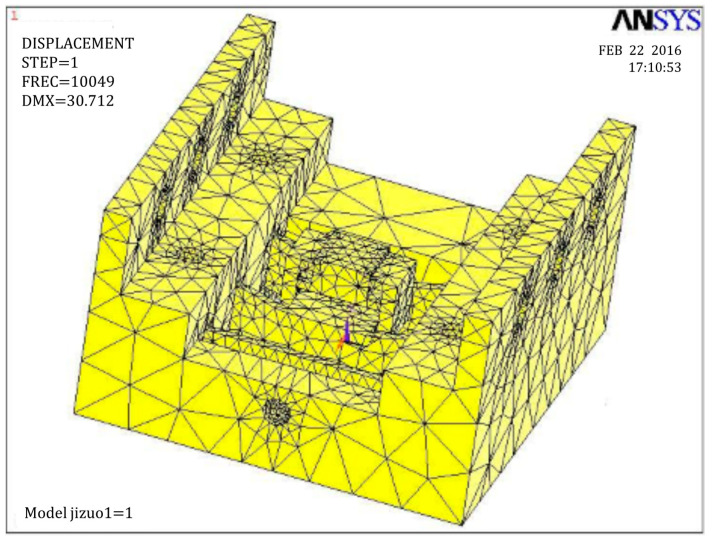
First order natural frequency of the stage base.

**Figure 6 micromachines-11-00765-f006:**
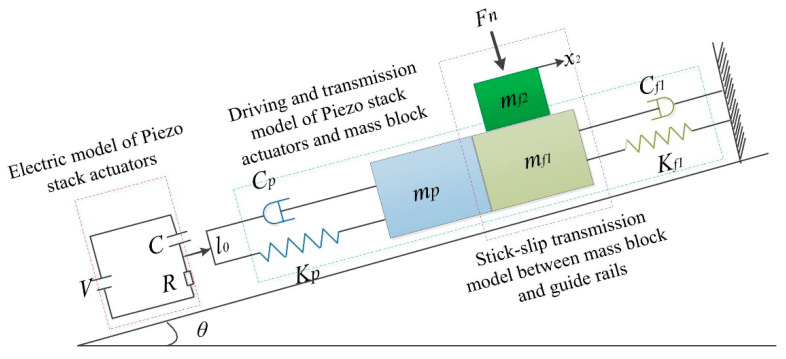
Dynamic model of the PASSNS.

**Figure 7 micromachines-11-00765-f007:**
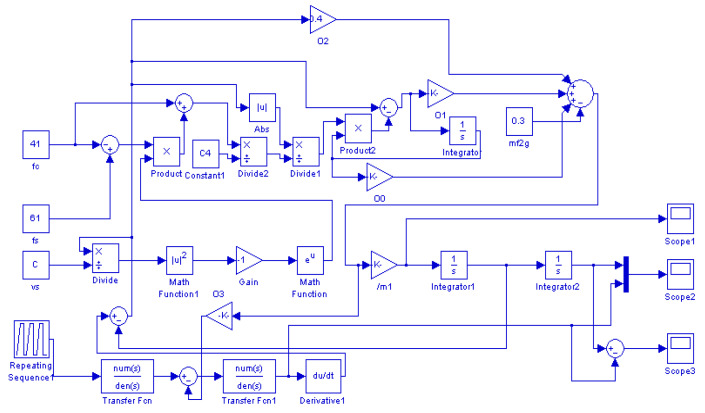
Simulation model the PASSNS. Reproduced with permission from [31].

**Figure 8 micromachines-11-00765-f008:**
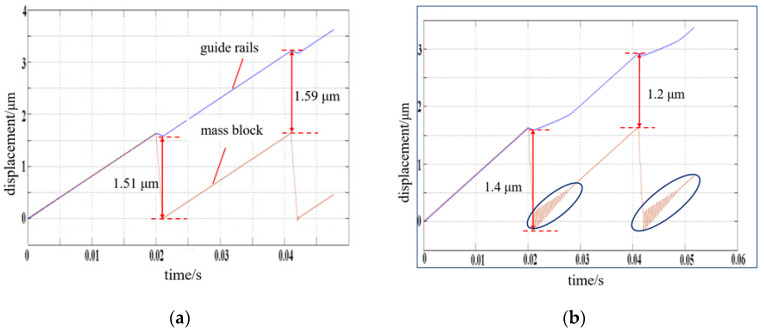
Simulation results of displacement under different total mass of adjusting unit and mass block: (**a**) total mass of 0.025 kg; (**b**) total mass of 0.25 kg.

**Figure 9 micromachines-11-00765-f009:**
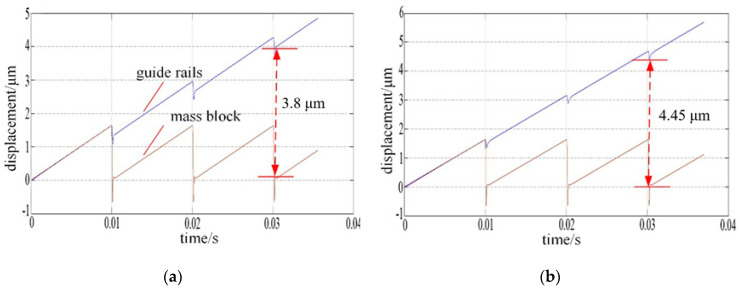
Simulation results of displacement under different working load: (**a**) 0.032 kg; (**b**) 0.064 kg.

**Figure 10 micromachines-11-00765-f010:**
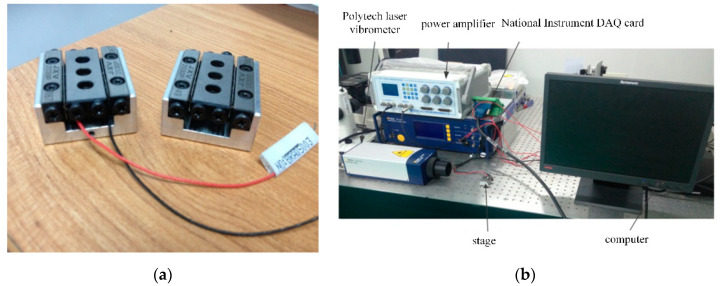
(**a**) Constructed stage. Reproduced with permission from [31]; (**b**) experimental system for performance measurement.

**Figure 11 micromachines-11-00765-f011:**
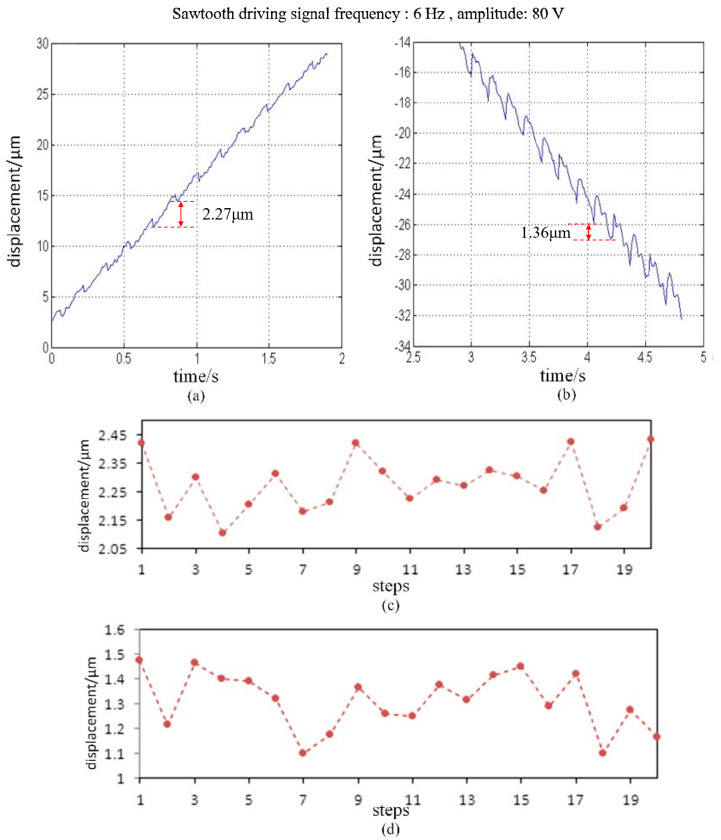
Forward and backward movement of the PASNS under a conventional sawtooth driving signal: (**a**) forward movement; (**b**) backward movement; (**c**) 20 continuous forward step displacements; and (**d**) 20 continuous backward step displacements.

**Figure 12 micromachines-11-00765-f012:**
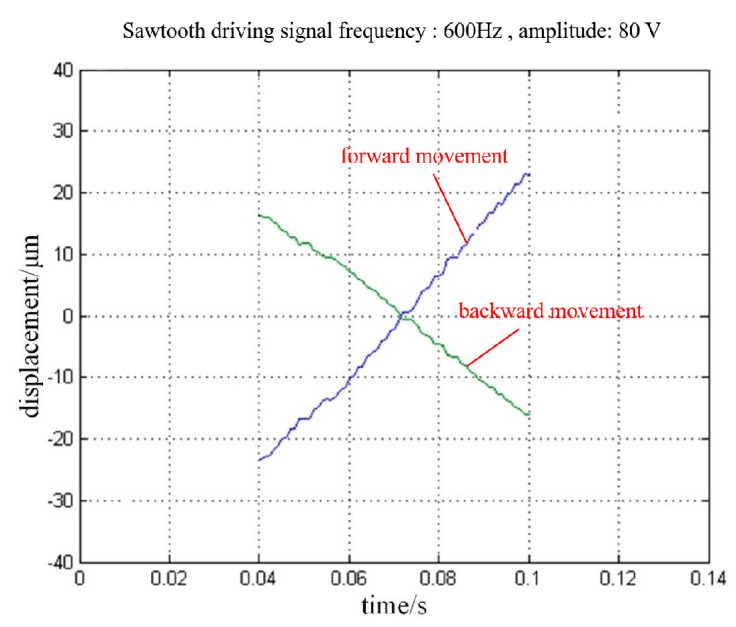
Forward and backward displacements of the PASSNS under the driving signal.

**Figure 13 micromachines-11-00765-f013:**
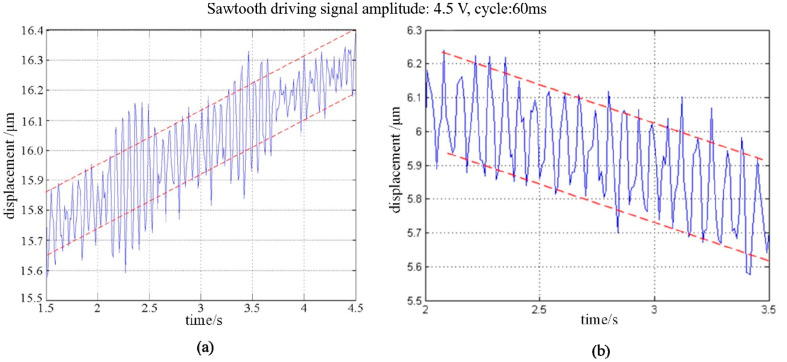
Forward and backward displacement of the stage under conventional sawtooth driving signal: (**a**) forward movement; (**b**) backward movement.

**Figure 14 micromachines-11-00765-f014:**
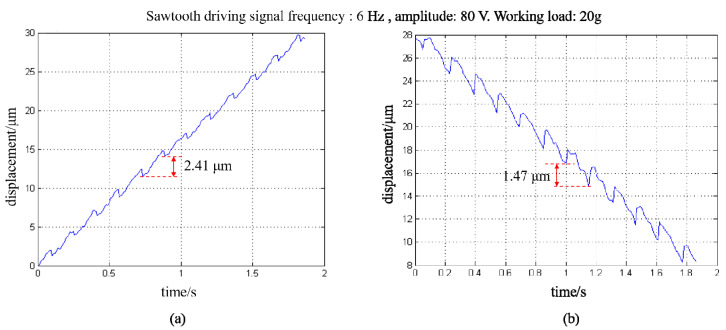
Forward and backward displacement of the stage with working load of 20 g: (**a**) forward movement; (**b**) backward movement.

**Figure 15 micromachines-11-00765-f015:**
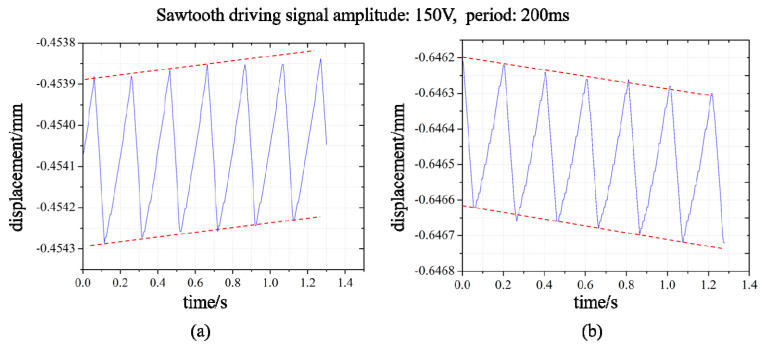
Displacement curves of the stage with diferent vertical loads: (**a**) vertical load of 72 g; (**b**) vertical load of more than 72 g.

**Table 1 micromachines-11-00765-t001:** Main characteristics of the developed stages of previous studies.

Researchers/Commercial Company	Main Characteristics of the Developed Stages				
	Dimension & Weight	Travel Range	Resolution	Maximum Speed	Maximum Output Force
Zhang et.al 2006	-	-	26 nm, 0.019°	7.2 mm/s	2.09 N
Rakotondrade et al.	20 × 20 × 20 mm^3^	50 mm, 360°	70 nm, 0.001°	1.8 mm/s, 20°/s	150 mN
Shimizu et al.	24 × 24 × 5 mm^3^	2 mm	10 nm	5 mm/s	60 mN
SmarAct GmbH (SLC-1720)	22 × 17 × 8.5 mm^3^, 13 g	12 mm	1 nm (OL) 50 nm (CL)	>20 mm/s	10 N
Attocube Inc. (ANPx101)	24 × 24 × 11 mm^3^, 20 g	5 mm	200 nm	3 mm/s	1 N

**Table 2 micromachines-11-00765-t002:** Characteristic parameters of the A, B, C hinges. FEA = finite element analysis.

Hinges	*L* (mm)	*h* (mm)	*T* (mm)	Theoretical Stiffness (N/μm)	FEA Simulation Stiffness (N/μm)	Maximum Stress (Mpa)
A	14	4	0.6	N/A	N/A	N/A
B	4	4	0.6	3.888	3.777	36.2
C	1.5	4	0.4	10.771	10.652	351.5

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
