# Peer review of "A Novel Stick-Slip Nanopositioning Stage Integrated with a Flexure Hinge-Based Friction Force Adjusting Structure"

_micromachines, 2020, doi:10.3390/mi11080765_

Round 1
Reviewer 1 Report
The section on Discussions and conclusions - some editing can make things clearer.
"In addition, some friction force unit is kind of heavy which result in the drastic vibration of stage during movement."
"Kind of heavy" is not the best way to write it - please rewrite.
Author Response
Thank you for your comments. In the beginning of the section on Discussion and Conclusion, the authors want to emphasize again the importance of friction force and the influence of the friction force adjusting units on the positioning performance of the PASSNS. In the previous description, the sentence “In addition, some friction force unit is kind of heavy which result in the drastic vibration of stage during movement.” is indeed not clear and precise. What we want to express is that the weight and size of the friction adjustment units will affect the motion performance of the positioning stage, especially the dynamic performance of the stage during movement.
In the revised manuscript, we rephrased some descriptions in the section on Discussion and Conclusion.
[Line 318-341]: As known to all, friction force plays an important role in the successful movement of the PASSNS. To effectively change the friction force, many friction force adjusting units have been integrated into the PASSNS. However, most of these unit cannot allow the stage to be robust to varying load. When the working load changes, the performance of stage will change drastically. In addition, some friction force adjusting units are heavy and large size, which result in the drastic vibration of stage during movement. To overcome these problems, we have developed a novel PASSNS where the flexure hinge-based friction force adjusting unit was employed to meet requirement. Numerical analysis was conducted to estimate the static performance of the PASSNS. To study the effect of different parameters on movement performance of the PASSNS, a dynamic model was established and simulation analysis was performed. Based on the simulation analysis, we found that different total mass of adjusting unit and mass block will affect the dynamic performance of the stage. We also found that in the conventional stage design, the step displacement of stage will become large with the increase of working load. The employment of flexure hinge-based friction force adjusting unit help us reduce the weight of adjusting unit, which improve the dynamic performance of the PASSNS. In addition, the deformation of flexure hinge can maintain the same when load changes. This ensure that stage can be robust to the varying load. This is further demonstrated by the experimental results. The maximum speed of forward and backward movement of stage has been measured to be 1.00 mm/s and 0.70 mm/s when the amplitude of driving signal was set to be 80 V. The minimum forward step displacement is around 11.00 nm, and the minimum backward step displacement is around 12.00 nm. The maximum vertical direction load capacity of stage is 300 g with robustness to varying load. As the proposed PASSNS provides many excellent performances, which demonstrate the great potential of the developed stage in many applications where require nanometer positioning accuracies, high travel stroke, compact structure, and large load.

Reviewer 2 Report
See comments on pdf file

Author Response
Dear Reviewer,
Thank you very much for the positive and constructive comments.
We have carefully addressed all the comments in the following responses and revised the manuscript accordingly. Please see the attachment.

This manuscript is a resubmission of an earlier submission. The following is a list of the peer review reports and author responses from that submission.
Round 1
Reviewer 1 Report
This work described a stick-slip nanopositioning stage with the flexure hinge-based friction force adjusting unit. This stage achieved robust performance to load change; in addition, it can move with a maximum speed of 1 mm/s (forward) and 0.7 mm/s (backward), and loading capacity of ~300 g. This work included CAD, theoretical calculation, simulation, and prototype devices. While a comprehensive study is presented in this work, I have the following comments.
1) Since nanopositioning stage is not a new concept, this work is expected to make breakthrough in, at least, one aspect of current devices. What is the major breakthrough? If the robustness is the major advantage over past devices, please describe clearly what are the shortcoming of previous devices.
2) At the beginning of introduction, the authors described the limitation of the macro-micro dual drive principle stage, i.e. bigger size. However, I do not think that compact size is the unique advantages of this work, please consider to revise the introduction.
3) The loading capacity lacks experimental confirmation. "(Line 241) As for the load capacity of the PASSNS in vertical direction, it depends on the load it can take. Based on the experimental results, we found if the load was less than 300 g, the nanopositioning stage was able to move successfully." This was not enough.
4) How does the performance in term of speed, step displacement compared to commercial nano-positioning stages?
5) Figures 8,9,11,12,13 lack necessary quality for publication. Please revise.
6) For figures to demonstrate experimental results, what is the number of technical repeats? A mean with standard deviation is necessary to demonstrate data.
Author Response
Dear Reviewer,
Thank you very much for the positive and constructive comments.
We have carefully addressed all the comments in the following responses and revised the manuscript accordingly.
Point 1: Since nanopositioning stage is not a new concept, this work is expected to make breakthrough in, at least, one aspect of current devices. What is the major breakthrough? If the robustness is the major advantage over past devices, please describe clearly what are the shortcoming of previous devices.
Response 1:
Thank you for your comments. Nanopositioning stage, especially the one driven by piezoelectric ceramics, has been studied by scholars for many years. And commercial nanopositioning stage has emerged, for example, PI, Newport. The piezoelectrically-actuated stick-slip nanopositioning stage (PASSNS) is a special nano positioning stage, which has the ability of long travel and cross-scale nanopositioning. While the friction force between the stick-slip surfaces plays a critical role in successful movement of stage, which influences the load capacity, dynamic performance, positioning accuracy. Many previous reported PASSNS are more focusing on stage structure design. Although there are literature reports on friction adjustment mechanisms and methods, such as mechanical tolerance fit, permanent magnet preloading, upper and lower springs preloading, but these friction adjustment methods are more complex, which will restrict the friction adjustment. At the same time, these friction adjustment methods have higher requirements on the machining accuracy of friction interface, which leads to higher requirements on the machining and assembly accuracy of PASSNS, and ultimately affects the positioning performance of the stage. This work is aimed at proposing a new friction force adjusting structure based on flexure hinge for the PASSNS to solve the friction force influence. The flexure hinge-based friction force adjusting structure is made by integrated manufacturing, the structure and adjustment method are simple, and easy to manufacture and assemble, and the assembly error is reduced, which make the proposed PASSNS be able to get good positioning performance and strong robustness to load change, and experiments results demonstrate the validity of the design.
In the revised manuscript, we have also added some descriptions in the introduction part to illustrate this issue.
[Line 71- 93]: …A systematic investigation and performance comparisons of different stick-slip and slip-slip modes of operation were previously discussed in Hunstig M’s research . Criteria such as steady state velocity, smoothness of motion, and start-up time were used for comparisons. Shimizu et al. utilized the stick-slip principle and piezoelectric actuators to develop an XY positioning stage. The stage is capable of moving over a range of ±1 mm in both directions at a travel speed of 5 mm/sec. In addition to the extensive research on PASSNS by scholars, some companies have issued related commercialized products, For example, the 1-DOF linear positioning stage (Model SLC-1720), a classic product of SmarAct GmbH, is designed based on the stick slip driving principle, its maximum stroke is 13mm, the maximum resolution is 50nm, the speed is 13mm / s, the weight is only 13g, and the size is 22 × 17 × 8.5mm3 . The most representative 1-DOF stick slip drive positioning stage of Attocube System Inc. is ANPX101 with size of 24 × 24 × 11mm3 and weight of 20g, its minimum step is 50nm, the stroke is 5mm, the speed can reach 3mm / s, and the maximum load can reach 1N.
The PASSNS reported in these literatures and some commercialized products are more about the structural design and positioning performance of the stage itself. Although there are literature reports on friction adjustment mechanisms and methods, such as mechanical tolerance fit, permanent magnet preloading , upper and lower springs preloading , but these friction adjustment methods are more complex, which will restrict the friction adjustment. At the same time, these friction adjustment methods have higher requirements on the machining accuracy of friction interface, which leads to higher requirements on the machining and assembly accuracy of PASSNS, and ultimately affects the positioning performance of the stage. Although many PASSNS have been developed, there still exist many problems such as low working load, step displacement sensitive to the load change, hard to obtain vertical movement, poor dynamic performance, and so on.
Point 2: At the beginning of introduction, the authors described the limitation of the macro-micro dual drive principle stage, i.e. bigger size. However, I do not think that compact size is the unique advantages of this work, please consider to revise the introduction.
Response 2: This comment is appreciated. Yes, compact size is not the unique advantages of this work. This work’s research is related to the stick-slip nanopositioning stage. As described in introduction [Line 51-53], Compared with them, stages developed by stick-slip drive principle not only have a compact structure, but also provide high resolution and long travel range.
Point 3: The loading capacity lacks experimental confirmation. "(Line 241) As for the load capacity of the PASSNS in vertical direction, it depends on the load it can take. Based on the experimental results, we found if the load was less than 300 g, the nanopositioning stage was able to move successfully." This was not enough.
Response 3: Thank you for the comment. The loading capacity of the PASSNS in vertical direction depends on the friction force between the stick-slip surfaces and slider mass. On the other hand, it depends on the load it can move upward. We have carried out the vertical load experiments, we apply a driving signal with a period of 200ms and an amplitude of 60V to the piezoelectric ceramic of the positioning stage. When the load is gradually increased in the vertical direction, and when it is increased to 300g, the positioning stage can move upward and output a rising displacement, as shown in Figure. 15(a). When the load is increased to more than 300g, the slider of the positioning stage does not move upward but fall down, as shown in Figure 15(b). Thus, based on the experimental results, the maximum load of stage in vertical direction is 300 g.
In the revised manuscript, we have added some descriptions in the last of the experimental result part to illustrate this issue.
Point 4: How does the performance in term of speed, step displacement compared to commercial nano-positioning stages?
Response 4: According to the experimental results, and compare to products with similar technology, for example, SLC-1720 from SmarAct GmbH, is capable of step width of 50-1500nm and max speed of more than 20mm/s. While the proposed stage in this work, the minimum forward and backward step displacements are approximately 11 nm and 12 nm, the maximum forward and backward movement speeds of the stage are 1 mm/s and 0.7 mm/s. Thus, our step displacement shows an advantage, but speed is less than SmarAct, due to our driver need to further improve in future work.
Point 5: Figures 8,9,11,12,13 lack necessary quality for publication. Please revise.
Response 5: This comment is appreciated. We have replaced better quality figures in the revised manuscript.
Point 6: For figures to demonstrate experimental results, what is the number of technical repeats? A mean with standard deviation is necessary to demonstrate data.
Response 6: Thank you for the comment. In the experiments, displacements of the positioning stage were measured by a noncontact laser distance sensor (KEYENCE, LKG3000 Series) in vertical direction and Polytech laser vibrometer in horizontal direction. All the experimental data are the average results after at least three times of repeated tests, and all figures are generated based on mean data.
Reviewer 2 Report
This authors present the prototype of piezo-actuated nano-positioning system. This is one of precision methods to drive a programmable x-y stage. In order to extend the limited range of the actuator, they proposed the use of stick-slip principle using a simple flexure hinge. The authors have designed their prototype using FEA, simulated the results using MATLAB, built the prototype using commercial equipment, and tested the forward and backward motion at variable loads.
I could not recommend the current manuscript for publication because of the following reasons:
- Novelty: I could not see the new contribution from this work. The use of flexure hinge in stick-slip principle have been widely reported. Two of the most recent ones are given herein. The first is reference [13] that was published in this journal in 2017 (https://www.mdpi.com/2072-666X/8/5/150). The second is the work by Zhang et al. (https://ieeexplore.ieee.org/abstract/document/8464681) that was published in 2018. In both cases, their design of flexure hinge and the positioning systems are much more advanced than the one presented in this manuscript. Therefore, the authors need to convince the reviewers on their novelty.
- Standard of literary presentation: Whether this is by accident or intentional (see comment #1), the design of the flexure hinge has not been well presented. The authors are suggested to follow the template from ref [13] and Zhang et al. (2018) to improve their presentation.
- Method: The fabrication and test of the system are not documented well. Readers cannot understand Figure 10 and replicate this experiment with the current data being provided.
- Results: The central hypothesis of this work is that the proposed flexure hinge manage to extend the range of the positioner. I could not find any evidence in the result to prove this. Some sort of comparison, for example with the system without the hinge etc. is warranted.
- Figure: (i) Figures 8, 11, and 13 are quite blurry, please replace with better quality. (ii) Figure 14 is incomplete. There is no label for axis and the delta y (average step?). (iii) Can the authors plot the graphs that display the varying frequency (Figure 12).
In closing, I am happy to review the revised version once the above concerns have been addressed.
Author Response
Dear Reviewer,
Thank you very much for the positive and constructive comments.
We have carefully addressed all the comments in the following responses and revised the manuscript accordingly.
Point 1: Novelty: I could not see the new contribution from this work. The use of flexure hinge in stick-slip principle have been widely reported. Two of the most recent ones are given herein. The first is reference [13] that was published in this journal in 2017 (https://www.mdpi.com/2072-666X/8/5/150). The second is the work by Zhang et al. (https://ieeexplore.ieee.org/abstract/document/8464681) that was published in 2018. In both cases, their design of flexure hinge and the positioning systems are much more advanced than the one presented in this manuscript. Therefore, the authors need to convince the reviewers on their novelty.
Response 1: Thank you for the comment. We agree with the reviewer that the usage of flexure hinge in stick-slip principle positioning stage have been widely reported, due to flexure hinges provide many advantages such as being monolithic with the rest of the mechanism, no backlash, and compactness. While the authors think the novel and breakthrough of this work are that we proposed a new friction force adjusting structure based on flexure hinge for the PASSNS to solve the friction force influence, helping reduce the weight of adjusting unit, which improve the dynamic performance of the PASSNS. In addition, the deformation of flexure hinge can maintain the same when load changes. This ensure that the stage get good positioning performance and strong robustness to load change.
Point 2: Standard of literary presentation: Whether this is by accident or intentional (see comment #1), the design of the flexure hinge has not been well presented. The authors are suggested to follow the template from ref [13] and Zhang et al. (2018) to improve their presentation.
Response 2: Thank you for the comments. In the revised manuscript, we have adjusted the order of introduction for the design of the proposed positioning stage, and the design of flexure hinge- based friction force adjustment structure is rephrased in section 2.1 & 2.2.
[Line 132-142] As mentioned above, friction force is the key to successful movement of PASSNS. To enable the stick-slip nanopositioning stage to move in both horizontal and vertical directions, it is necessary to adjust the friction force between the driving object and slider. As the flexure hinges provide many advantages over conventional joints such as being monolithic with the rest of the mechanism, no need for lubrication, no backlash, and compactness, it is employed to uniformly adjust friction force between driving object and slider with a bolt in contact with its center. Schematic of the flexure hinge structure C in Figure 2, which is used to adjust friction force, is shown in Figure 3. This structure consists of two symmetric corner-filleted flexure hinges. When the bolt for adjusting friction force is screwed upwards, flexure hinges C will deform symmetrically as shown in Figure 3 (b). Finally, the friction force between these two parts will increase accordingly. The flexure hinge-based friction force adjusting structure is made by integrated manufacturing, the structure and adjustment method are simple, and easy to manufacture and assemble, and the assembly error is reduced.
Point 3: Method: The fabrication and test of the system are not documented well. Readers cannot understand Figure 10 and replicate this experiment with the current data being provided.
Response 3: Thank you for the comments. We have revised the descriptions of Figure 10 for the fabrication and test of the system in section 3. And some figures of experimental results are also revised, such as Figure 8, Figure 11, Figure 14, Figure 15.
Point 4: Results: The central hypothesis of this work is that the proposed flexure hinge manage to extend the range of the positioner. I could not find any evidence in the result to prove this. Some sort of comparison, for example with the system without the hinge etc. is warranted.
Response 4: Thank you for the comment. This work is aimed at proposing a new friction force adjusting structure based on flexure hinge for the PASSNS to solve the friction force influence. The flexure hinge-based friction force adjusting structure is made by integrated manufacturing, the structure and adjustment method are simple, and easy to manufacture and assemble, and the assembly error is reduced, which make the proposed PASSNS be able to get good positioning performance and strong robustness to load change, and experiments results demonstrate the validity of the design.
Point 5: Figure: (i) Figures 8, 11, and 13 are quite blurry, please replace with better quality. (ii) Figure 14 is incomplete. There is no label for axis and the delta y (average step?). (iii) Can the authors plot the graphs that display the varying frequency (Figure 12).
Response 5: The comments are appreciated. We have replaced better quality figures in the revised manuscript. For Figure 14, we felt sorry for this mistake and we have corrected it in the revised manuscript too.
Reviewer 3 Report
The author presented the development and testing of a stick-slip nanoposition stage integrated with a friction force adjusting structure. As I think the work is not clearly presented and needs to be revised before accepted for publication:
1) In the background introduction part, the author reviewed some previously published work. However, some companies (Attocube and Smaract and etcs) have commercialized such stages. Their work should also be reviewd.
2) The design of the stage (section 2.1 & 2.2) was not clearly stated and need to be revised.
3) The testing section (lines 185-190) was not clearly stated an need to be revised. For example, in line 189 the author declared that he used KEYENCE LKG3000 (is this tool precise enough?) for testing. However, in figure 10b, he showed a Polytech laser vibrometer. I suggested the author revised this part, add a schematic view of test setting as well as a zoom in picture of the stages under test.
4) Finally, the author claimed a improved performance of this stage. However, compared with closed-loop stage with similar dimension (for example, Smaract SLC1720), is there any performance improvement? The author may need to carefully defined his rival pool at the the beginning of the paper, for example, comparing with an open-loop stage without the friction force adjusting structure. The closed-loop stages usually have a larger footprint than open-loop one.
Author Response
Dear Reviewer,
Thank you very much for the positive and constructive comments.
We have carefully addressed all the comments in the following responses and revised the manuscript accordingly.
Point 1: In the background introduction part, the author reviewed some previously published work. However, some companies (Attocube and Smaract and etcs) have commercialized such stages. Their work should also be reviewd.
Response 1: Thank you for the comment. In the revised manuscript, we have added some commercialized related stages technologies in the background introduction.
[Line 76-93] … In addition to the extensive research on PASSNS by scholars, some companies have issued related commercialized products, For example, the 1-DOF linear positioning stage (Model SLC-1720), a classic product of SmarAct GmbH, is designed based on the stick slip driving principle, its maximum stroke is 13mm, the maximum resolution is 50nm, the speed is 13mm / s, the weight is only 13g, and the size is 22 × 17 × 8.5mm3 . The most representative 1-DOF stick slip drive positioning stage of Attocube System Inc. is ANPX101 with size of 24 × 24 × 11mm3 and weight of 20g, its minimum step is 50nm, the stroke is 5mm, the speed can reach 3mm / s, and the maximum load can reach 1N.
The PASSNS reported in these literatures and some commercialized products are more about the structural design and positioning performance of the stage itself. Although there are literature reports on friction adjustment mechanisms and methods, such as mechanical tolerance fit, permanent magnet preloading , upper and lower springs preloading , but these friction adjustment methods are more complex, which will restrict the friction adjustment. At the same time, these friction adjustment methods have higher requirements on the machining accuracy of friction interface, which leads to higher requirements on the machining and assembly accuracy of PASSNS, and ultimately affects the positioning performance of the stage. Although many PASSNS have been developed, there still exist many problems such as low working load, step displacement sensitive to the load change, hard to obtain vertical movement, poor dynamic performance, and so on.
Point 2: The design of the stage (section 2.1 & 2.2) was not clearly stated and need to be revised.
Response 2: Thank you for the comments. In the revised manuscript, we have adjusted the order of introduction for the design of the proposed positioning stage, and the design of flexure hinge- based friction force adjustment structure is rephrased.
[Line 132-148] As mentioned above, friction force is the key to successful movement of PASSNS. To enable the stick-slip nanopositioning stage to move in both horizontal and vertical directions, it is necessary to adjust the friction force between the driving object and slider. As the flexure hinges provide many advantages over conventional joints such as being monolithic with the rest of the mechanism, no need for lubrication, no backlash, and compactness, it is employed to uniformly adjust friction force between driving object and slider with a bolt in contact with its center. Schematic of the flexure hinge structure C in Figure 2, which is used to adjust friction force, is shown in Figure 3. This structure consists of two symmetric corner-filleted flexure hinges. When the bolt for adjusting friction force is screwed upwards, flexure hinges C will deform symmetrically as shown in Figure 3 (b). Finally, the friction force between these two parts will increase accordingly. The flexure hinge-based friction force adjusting structure is made by integrated manufacturing, the structure and adjustment method are simple, and easy to manufacture and assemble, and the assembly error is reduced.
Point 3: The testing section (lines 185-190) was not clearly stated an need to be revised. For example, in line 189 the author declared that he used KEYENCE LKG3000 (is this tool precise enough?) for testing. However, in figure 10b, he showed a Polytech laser vibrometer. I suggested the author revised this part, add a schematic view of test setting as well as a zoom in picture of the stages under test.
Response 3: The comment is much appreciated. Both Polytech laser vibrometer and KEYENCE LKG3000 are used in our experiment. Because the Polytec laser vibrometer is more suitable for measuring the horizontal direction, therefore, KEYENCE laser micrometer is used to measure the vertical movement of the positioning stage. Figure 10 (b) shows horizontal direction movement measured by Polytec laser vibrometer.
we have rephrased some statements of Figure 10 in revised manuscript.
Point 4: Finally, the author claimed an improved performance of this stage. However, compared with closed-loop stage with similar dimension (for example, Smaract SLC1720), is there any performance improvement? The author may need to carefully defined his rival pool at the beginning of the paper, for example, comparing with an open-loop stage without the friction force adjusting structure. The closed-loop stages usually have a larger footprint than open-loop one.
Response 4: The comment is much appreciated. As the first comment, we have added some descriptions and comparisons of rivals to highlight the advantages of the proposed stage. In this work, the employment of flexure hinge-based friction force adjusting unit help us reduce the weight of adjusting unit, which improve the dynamic performance of the PASSNS. In addition, the deformation of flexure hinge can maintain the same when load changes. This ensure that stage can be robust to the varying load.
Round 2
Reviewer 1 Report
The authors well addressed my comments in the revised version of manuscript. I would suggest to accept this work as is.
Reviewer 3 Report
I have no more questions.